# Phenotypic Variation of Autosomal Recessive Leber Hereditary Optic Neuropathy (arLHON) in One Family

**DOI:** 10.3390/diagnostics12112701

**Published:** 2022-11-05

**Authors:** Dorota Pojda-Wilczek, Justyna Wójcik, Bożena Kmak, Maciej Robert Krawczyński

**Affiliations:** 1Department of Ophthalmology, Faculty of Medical Sciences in Katowice, Medical University of Silesia in Katowice, 40-055 Katowice, Poland; 2Kornel Gibiński University Clinical Centre, 40-514 Katowice, Poland; 3Individual Medical Practice Justyna Wójcik, 32-500 Chrzanów, Poland; 4Students’ Scientific Society, Department of Ophthalmology, Faculty of Medical Sciences in Katowice, Medical University of Silesia in Katowice, 40-055 Katowice, Poland; 5Department of Medical Genetics, Poznan University of Medical Sciences, 61-701 Poznań, Poland; 6Center for Medical Genetics GENESIS, 60-406 Poznań, Poland

**Keywords:** Leber hereditary optic neuropathy, LHON, arLHON, idebenone, Photopic Negative Response ERG, Visual Evoked Potentials

## Abstract

Leber hereditary optic neuropathy (LHON) is a rare disease with a prevalence of 1 in 25,000 births. LHON usually presents in young males, with painless loss of visual acuity in one or both eyes. Recently an autosomal recessive form of the disease (arLHON or LHONAR) has been described, which is caused by a biallelic mutation in the *DNAJC30* gene (usually a missense mutation c.152A>G). The phenotypic and clinical characteristics of patients with arLHON are similar to those of mtLHON, but some differences have been described. Therapy is problematic and challenging. This paper describes clinical and electrophysiological findings in one family (three children and two parents) with arLHON and emphasizes the role of Photopic Negative Response Electroretinography, which provides objective measurement of retinal ganglion cells function. In Leber hereditary optic neuropathy, abnormal retinal ganglion cells function can be found in both eyes, even if visual acuity loss only occurs in one eye. Early clinical diagnosis, confirmed by genetic analysis, may be the key to sight-preserving treatment.

## 1. Introduction

In 1871, German ophthalmologist Theodor Leber first described the disease as characterized by the bilateral subacute loss of central vision, resulting from degeneration of retinal ganglion cells and the optic nerve. Leber hereditary optic neuropathy (LHON) is a rare disease with a prevalence of 1 in 25,000 births and is more common in males. It is maternally-inherited, and is usually caused by one of three mtDNA mutations, m.3460G>A, m.11778G>A, or m.14484T>C. LHON, and mainly affects young adults, in their second to third decade of life, but symptoms can also be seen in childhood. In the acute phase, visual acuity loss and a centrocecal scotoma occur. Other clinical features include impairment of color perception. Unlike in optic neuritis, pupillary reflexes are preserved, and patients usually do not report pain associated with eye movement. After 6 months, the retinal nerve fiber layer gradually degenerates, leading to atrophy of the optic nerve. Several factors can induce or exacerbate LHON, such as smoking or alcohol consumption, as they increase oxidative stress.

Recently an autosomal recessive form of the disease (arLHON or LHONAR) has been described. It is caused by a biallelic mutation in the *DNAJC30* gene (usually a missense mutation c.152A>G). The phenotypic and clinical characteristics of patients with arLHON are similar to those of mtLHON, but some differences have been described. Therapy is problematic and challenging [1,2,3].

The three mtDNA mutations lead to complex I deficiency, a common mitochondrial respiratory chain defect. The *DNAJC30*, on the other hand, is a chaperone protein facilitating adenosine triphosphate (ATP) synthesis dependent on complex I and complex V in the mitochondrial respiratory chain. Low ATP synthesis along with high levels of reactive oxygen species might be involved in the degeneration of retinal ganglion cells [3].

## 2. Case Study

Our study concerns one family consisting of: the eldest child, a girl; the middle and the youngest child, both boys, five and seven years younger than their sister; the parents (both 46 years old). The parents observed a sudden loss of visual acuity in the eldest child when she was 7 years old. Her far visual acuity (FVA) was 2/25 in the right eye (RE) and 5/16 in left eye (LE) (Snellen charts, metric scale); near visual acuity (NVA) was 2.0 (RE) and 1.5 (LE). Abnormal color vision was also found. Optic neuritis was suspected. Brain MRI (magnetic resonance imaging) revealed a small amount of fluid in the optic nerve sheath. The girl was also observed for epilepsy. Neurological and ophthalmic examinations did not reveal the cause of vision impairment. Idiopathic optic atrophy was diagnosed. One year later, FVA of RE and LE were 0.2 and NVA was 1.5, without correction. The refractive error was +1.0 D, but correction did not improve VA. Mild alternating exotropia was evident. The temporal sectors of the optic discs were pale. In the next two years, the girl developed bilateral rotary nystagmus, and, in the subsequent two years, she was diagnosed with LE myopia (−2.75 D). The myopic shift in the LE may only have resulted from accommodative effort. Due to exotropia the patient started using her RE for far vision and LE for near vision. When the girl was fifteen, a refraction test (no cycloplegics) revealed her RE was still hyperopic while her LE accommodated to −4 D; a cycloplegic refraction test showed as follows: RE +0.75 D, LE −3.0 D. At seventeen, a refraction test revealed −5 D (LE); the results of cycloplegic refraction were the same.

The boys had their first ophthalmic examination when they were 5 and 3 years old. The middle child was hyperopic (+1.5 D); uncorrected FVA was 4/6 in both eyes. In the following year, uniocular FVA improved to 4/5, and binocular FVA to 4/4. The youngest child was diagnosed with the anisometropic amblyopia of his RE with FVA of 4/10 (refractive error +2 D); his LE FVA was 4/5 (refractive error +0.5 D). A RE corrective lens (+1.0 D) and LE patching for several hours a day were recommended. The boy received no treatment and there was no follow-up.

At the age of 10, the middle child complained of blurred vision while reading, but no abnormalities were found on ophthalmic examination; VA was still normal.

At the age of 12, the middle child underwent routine visual acuity examination at school, which revealed poor vision in one eye. Ophthalmic examination revealed RE FVA 5/5 and LE FVA 1/50. Color vision (Ishihara Test) was normal in the RE and absent in the LE. Pattern Visual Evoked Potentials (PVEP) (EP-1000 Tomey, Japan, ISCEV Standard [4]) demonstrated prolonged P100 latencies in both eyes and low amplitudes in left eye (Table 1). Flash Visual Evoked Potentials (FVEP) were normal (Table 1).

Full-field flash Electroretinography (ffERG) was performed with the RETeval portable unit (LKC, USA) and sensor strip electrodes, in accordance with the ISCEV ERG standard [5]. The scotopic full-field ERG (ffERG) was normal while photopic responses were subnormal. The a-wave and b-wave amplitudes and implicit times of photopic ffERG are presented in Table 2.

The Photopic Negative Response ERG (PhNRERG) indicated very low activity in retinal ganglion cells in both eyes (Table 3).

Visual field (Octopus 1-2-3, glaucomatous program tG1; Interzeag, Switzerland) was normal in the RE and central absolute scotoma was found in the LE (Figure 1).

Optic neuritis was suspected, but an MRI scan showed no abnormalities; there was also no response to treatment with intravenous steroids. Multiple sclerosis, brain tumor, and encephalitis were considered in the differential diagnosis, but were ruled out by imaging tests. Considering retinal ganglion cells abnormalities, no improvement in visual acuity in the left eye and decreased visual acuity in the right eye, a decision was made to perform genetic testing. In the same period, far RE visual acuity decreased to 5/50, and LE was still 1/50. The boy was unable to read whole words; however, he was able to read large letters (NVA 2.0). VEP and ERG tests, on the other hand, showed improvement in LE function (Table 1 and Table 3). DNA for genetic testing was extracted from the patient’s .peripheral blood lymphocytes. After ruling out three main LHON mtDNA mutations, a large fragment of exon 1 of *DNAJC30* gene was amplified and sequenced (Sanger sequencing on ABI 3130xl Genetic Analyser using Applied Biosystems Big Dye Terminator v3.1 Cycle sequencing kit). The sequenced amplicon captures three main pathogenic variants of *DNAJC30*, i.e., c.152A>G, c.232C>T, and c.302T>A.

The homozygous pathogenic variant c.152A>G (p.Tyr51Cys) was identified in the middle child and in his older sister. Segmentation analysis confirmed that both parents were carriers of the heterozygous variant. This variant was also found in the youngest child.

Two months after the first signs of neuropathy (early April), treatment with idebenone at an initial dose of 800 mg per day was started, resulting in further improvement in retinal ganglion cells function (PhNRERG) and better VEP results. After four months of treatment, PhNRERG was within normal limits in both the left and right eyes while visual acuity stabilized at 1/50 (m). There was a central scotoma in the visual field of both eyes (Figure 2).

The eldest child also began idebenone therapy, with similar results to the middle child, i.e., no change in visual acuity, but an improvement in retinal ganglion cell function was observed. The RE visual field (Kinetic Goldmann Perimeter) showed a small central scotoma (Figure 3).

Ophthalmic examinations performed in the parents showed no abnormalities in visual acuity, but the mother had an abnormal ffERG (Table 2 and Table 3). Color fundus imaging was performed using a fundus camera (Carl Zeiss Meditec Inc., Germany, Template Version 0.1). A similar fundus pattern was found in all children and their mother (Figure 4A–E).

The follow-up examination was performed in August, after four months of treatment. The results of electrophysiological testing are shown in Table 1 and Table 3. VEP was performed using the Reti-Port system (Roland Consult, Brandenburg an der Havel, Germany); ffERG was performed with a portable unit-RETeval (LKC Technologies, Gaithersburg, MD, USA). Ganglion Cells Complex layer (GCC) and Retinal Nerve Fiber Layer (RNFL) thickness measurements (Cirrus HD-OCT Carl Zeiss Meditec Inc., Jena, Germany) are collectively shown in Table 4.

## 3. Discussion

Leber hereditary optic neuropathy (LHON) is a mitochondrial disease. It is caused by point mutations in mitochondrial DNA leading to dysfunction of oxidative phosphorylation complex I, which mainly affects retinal ganglion cells (RGC). The main mutations, starting with the most common, are 11778G>A in the *MTND4* gene, 14484T>C in the *MTND6* gene, and 3460G>A in the *MTND1* gene. In clinical settings, it is impossible to distinguish these mutations. arLHON, an autosomal recessive form of Leber hereditary optic neuropathy, has been identified as being genetically determined by mutations in the *DNAJC30* gene, located within the cell nucleus. Compared to mtLHON, patients suffering from arLHON tend to be younger at the onset of their symptoms. A homozygous c.152A>G mutation (p.Tyr51Cys) in the DNAJC30 gene is found in 90% of arLHON cases. Furthermore, patients with a mutation in the *DNAJC30* gene show symptoms in both eyes and are mostly male; probands tend to have higher recovery rates. Kieninger et al. [3] have suggested 7.7% of patients diagnosed with LHON may suffer from *DNAJC30*-linked arLHON.

A typical homozygous pathogenic variant, i.e., the c.152A>G (p.Tyr51Cys), was identified in the above family. This variant is the most common cause of arLHON, including among the Polish population.

LHON usually manifests in young males, with painless loss of visual acuity in one or both eyes [1]. Most frequently, the disease begins with optic disc edema in one eye; visual acuity decreases in both eyes over the following weeks or months. In this particular family, the disease did not, initially, cause changes in optic disc morphology. Over time though, the disc became grayish, but not pale. Ganglion cell and retinal nerve fiber layers became thicker in family members with the homozygous compared to heterozygous variant, except for the mother whose RNFL thickness and retinal function (ffERG and PhNRERG) were similar to those found in her homozygous children. Interestingly, the low RGC activity found in the PhNRERG was not associated with low RGC thickness in OCT.

Idebenone treatment improves the function of retinal ganglion cells [7], which has been confirmed in our study. Although it cannot reverse the damage that has already occurred, i.e., loss of central vision, and the treatment should be implemented as soon as possible [8].

As stated above, LHON symptoms are due to a genetically determined ATP deficiency in retinal ganglion cells, namely, mutations targeting complex I in the mitochondrial respiratory chain. Idebenone is a synthetic analogue of coenzyme Q10, a short-chain benzoquinone. Nicotinamide adenine dinucleotide phosphate (NAD(P)H: quinone oxidoreductase 1) reduces the drug inside the cells to hydroquinone, a form which is capable of transporting electrons straight into complex III in the mitochondrion. Thus, the dysfunctional Complex I is passed by and the energy generation by RGCs is restored. Idebenone not only acts as an electron transporter, but also inhibits lipid peroxidation; it can, therefore, effectively protect mitochondria and cells in general from oxidative damage [9]. If some retinal ganglion cells are not irreversibly damaged, this treatment might result in visual acuity improvement.

Visual acuity depends on the size of central scotoma. The preserved peripheral visual field may indicate that the midget retinal ganglion cells (P-cells) need more energy than the parasol retinal ganglion cells (M-cells). Midget RGCs are involved in color discrimination, pattern, texture, and stereoscopic depth perception. This complex function requires a lot of energy. P-cells, on the other hand, are small with less mitochondrial reserve. In LOHN, this RGC type is damaged first and irreversibly, and this happened in our patients [10]. Timely diagnosis is very important as it may help avoid RGC atrophy. The PhNRERG revealed improvement of RGC function during treatment, but there was no improvement in visual acuity. It is quite likely that idebenone supports the function of those RGC that have not been irreversibly affected; therefore, this therapy may prevent blindness.

If a patient is aware that they carry the genetic mutations associated with LHON, they are advised to avoid smoking, as tobacco has been proven to be a factor leading to the development of symptoms [11]. Other lifestyle components that could trigger LHON symptoms have yet to be identified, leaving the probands with limited options for preventing the onset of the disease. Pfeffer et al. [12] have suggested dietary changes are beneficial in LHON and can be used as a treatment option, especially a reduced-calorie ketogenic diet. Since the pathophysiology of LHON is associated with abnormalities in the electron transport chain (ETC), researchers have turned to antioxidants, such as the B-group vitamins or folic acid, as a form of treatment [13]. However, insufficient amounts of data have been collected so far. The ideal treatment for Leber hereditary optic neuropathy would be gene therapy. Patients with the *MTND4* gene mutation are the most numerous and, therefore, constitute the research focus [14,15,16]. Consequently, gene therapy focuses on delivering a properly functioning *MTND4* gene to the mitochondria of the proband’s retinal ganglion cells. Due to the inconsistent clinical results, gene therapy is not widely used to treat LHON. Another treatment option currently under investigation is the use of quinone alpha-tocotrienol, but more evidence and research is needed to establish its validity [17].

Studies have shown that LHON is 3–7 times more common in men. However, the ratio is 1:1 in patients whose symptoms first occurred when they were younger than 5 or older than 45 years. A proband can develop symptoms at any point in life. While men usually start being symptomatic between the ages of 14 and 26, women show no clear pattern and develop symptoms at different ages [18].

## 4. Conclusions

Patients with Leber hereditary optic neuropathy may exhibit retinal ganglion cell dysfunction in both eyes even if the loss of visual acuity occurs in one eye only. Early diagnosis confirmed by genetic analysis may be the key to sight-preserving treatment. Rare mutations should also be considered in genetic analysis.

## Figures and Tables

**Figure 1 diagnostics-12-02701-f001:**
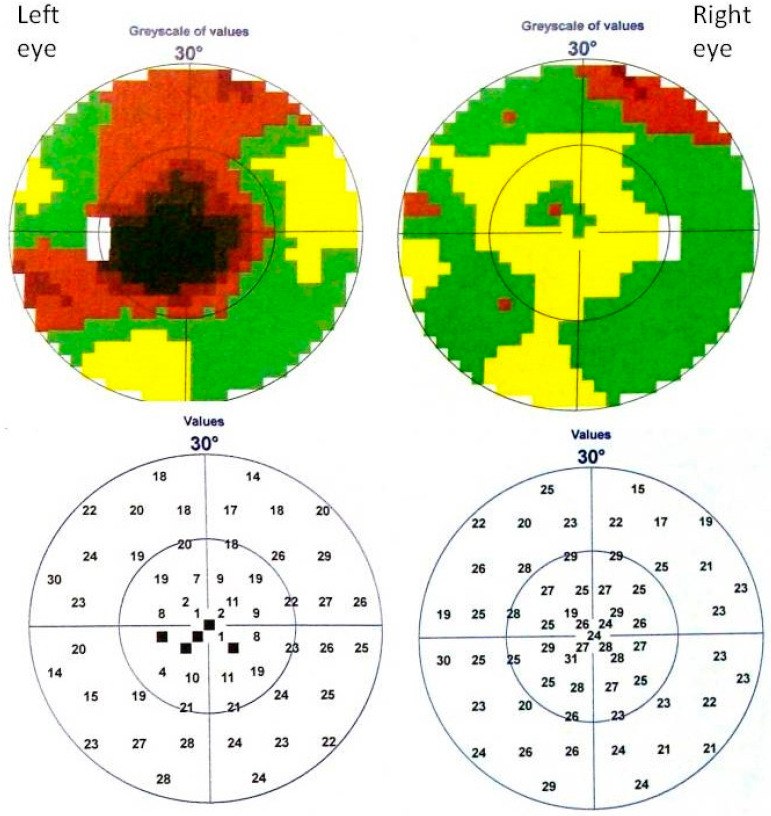
The middle child. First examination (January, the year of diagnosis). Absolute central scotoma was found in the visual field of the LE (Octopus 1-2-3, glaucomatous program tG1). Mean Defect: right eye 5.1, left eye: 12.9.

**Figure 2 diagnostics-12-02701-f002:**
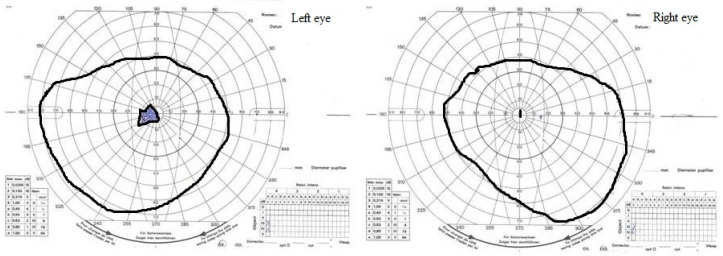
The middle child. Kinetic visual field examination (August). A central scotoma was in the visual field of both the left and right eye.

**Figure 3 diagnostics-12-02701-f003:**
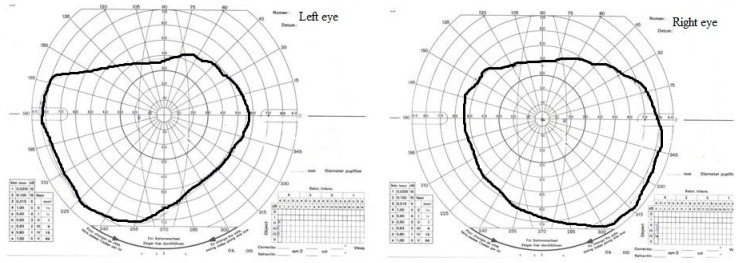
The eldest child. Kinetic visual field examination (August). A small central scotoma in the right-eye visual field.

**Figure 4 diagnostics-12-02701-f004:**
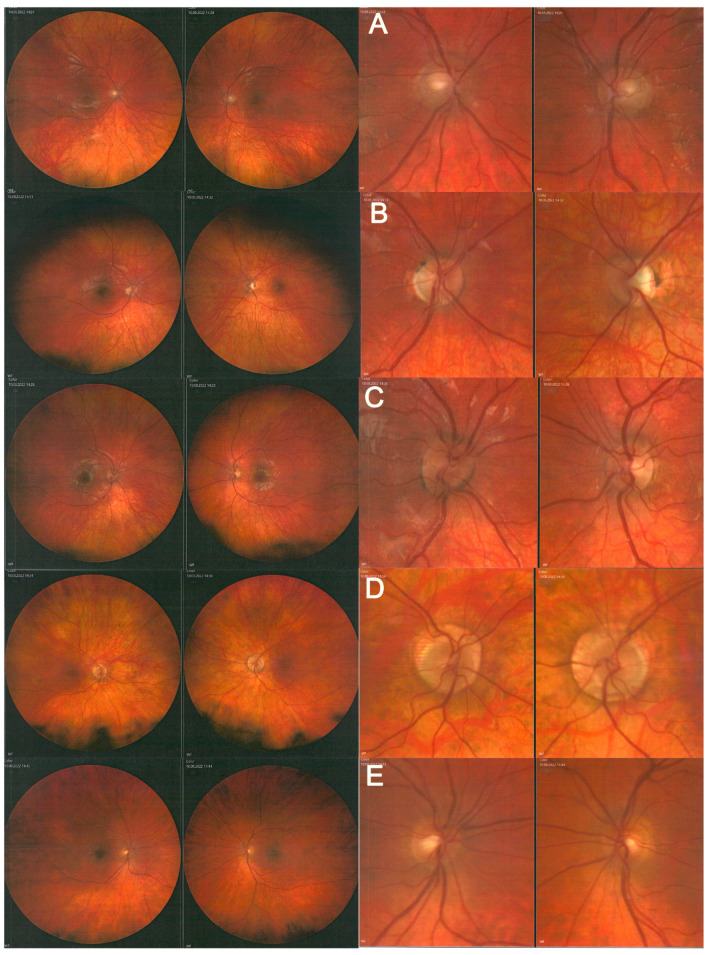
Eye fundus (left) and optic disc (right). (**A**)—the middle child; (**B**)—the eldest child; (**C**)—the youngest child; (**D**)—mother; (**E**)—father.

**Table 1 diagnostics-12-02701-t001:** Pattern and Flash Visual Evoked Potentials (VEP). The latency (L) and amplitude (A) of P100 waves and P2 waves. R—right eye; L—left eye; NM—non-measurable; NP—not performed.

PATIENT	Examination Date	EYE	Pattern VEP P100				Flash VEP P2	
			1°		15′		1.4 Hz	
			L [ms]	A [µV]	L [ms]	A [µV]	L [ms]	A [µV]
middle child 12-year-old	January	R	125	14.4	121	17.2	104	19.8
		L	164	5.2	NM	NM	115	18.2
	May	R	120	8.2	129	4.9	110	19.1
		L	145	4.9	NM	NM	107	14.2
	August	R	149	5.12	129	1.95	109	13.7
		L	145	4.09	102	1.95	106	10.3
eldest child	10-year-old	R	105	7.4	129	5.1	NP	NP
		L	109	7.4	128	8.9	NP	NP
	17-year-old	R	98	6.2	118	2.6	151	10.3
		L	103	6.3	115	2.9	149	11.5
youngest child	10-year-old	R	104	25.7	108	19.0	102	18.1
		L	113	28.5	118	20.8	107	31.0

**Table 2 diagnostics-12-02701-t002:** The photopic ffERG performed in August in the year of arLHON diagnosis. LA—light adapted; IT—implicit time; A—amplitude; R—right; L—left. Reference ranges according to LKC. Values in parentheses are percentiles.

PATIENT	EYE	LA 3 ERG				LA 30 Hz ERG	
		a-Wave		b-Wave		Peak	
		IT [ms]	A [µV]	IT [ms]	A [µV]	IT [ms]	A [µV]
Reference ranges children		9.8↔14.0	−2.9↔−16.8	25.3↔30.5	21.0↔68.6	23.2↔28.1	20.0↔57.1
middle child	R	12.9 (82%)	−9.0 (66%)	27.7 (47%)	39.3 (54%)	24.3 (40%)	36.7 (65%)
	L	13.1 (91%)	−5.2 (12%)	28.0 (63%)	30.2 (20%)	24.8 (62%)	25.1 (18%)
eldest child	R	13.0 (83%)	−3.3 (3%)	30.0 (95%)	25.3 (10%)	27.2 (96%)	26.9 (25%)
	L	10.9 (15%)	−3.5 (4%)	28.6 (74%)	23.8 (9%)	26.8 (95%)	23.7 (16%)
youngest child	R	11.8 (51%)	−9.6 (73%)	29.0 (87%)	43.4 (70%)	25.4 (86%)	48.1 (92%)
	L	11.9 (56%)	−8.8 (60%)	28.8 (83%)	38.9 (53%)	25.4 (87%)	43.6 (82%)
Reference ranges adults		6.6↔13.6	−1.2↔−18.5	24.0↔32.1	11.1↔72.6	23.4↔28.6	13.9↔67.4
mother	R	12.7 (76%)	−1.8 (4%)	27.9 (14%)	5.6 (1%)	24.7 (17%)	6.8 (0%)
	L	11.3 (34%)	−3.1 (8%)	28.3 (24%)	10.8 (2%)	25.2 (35%)	11.9 (1%)
father	R	11.1 (24%)	−4.3 (18%)	28.8 (36%)	27.7 (46%)	26.1 (70%)	28.4 (54%)
	L	10.9 (18%)	−2.8 (7%)	30.5 (83%)	23.0 (28%)	26.2 (74%)	21.8 (25%)

**Table 3 diagnostics-12-02701-t003:** Photopic Negative Response Electroretinography performed in the year of diagnosis. Photopic Negative wave implicit time (IT), amplitude (A), and W-ratio (W-ratio = (b-p_min_)/(b-a) [6]). Reference ranges are according to LKC. Values in parentheses are percentiles.

PATIENT	DATE	EYE	IT [ms]	A [µV]	W-Ratio
middle child	January	R	64 (29%)	15.1 (0%)	0.79 (0%)
		L	78 (69%)	2.5 (1%)	1.00 (4%)
	February	R	60 (21%)	−3.5 (27%)	1.08 (48%)
		L	61 (21%)	−4.7 (44%)	1.16 (69%)
	March	R	42 (14%)	−4.0 (35%)	0.90 (34%)
		L	46 (16%)	−3.5 (21%)	0.97 (74%)
	May	R	40 (13%)	−5.6 (80%)	0.96 (71%)
		L	43 (14%)	−4.1 (37%)	0.95 (65%)
	August	R	39 (21%)	−6.6 (87%)	1.11 (99%)
		L	39 (22%)	−5.3 (67%)	1.12 (100%)
eldest child	March	R	73 (92%)	−3.9 (34%)	0.83 (10%)
		L	75 (94%)	−3.0 (10%)	0.82 (8%)
	August	R	46 (33%)	−3.4 (18%)	0.93 (48%)
		L	44 (33%)	−2.8 (6%)	0.88 (20%)
youngest child	August	R	61 (74%)	−4.9 (59%)	0.98 (70%)
		L	59 (70%)	−4.8 (58%)	0.94 (48%)
mother	August	R	77 (95%)	−1.8 (1%)	0.77 (3%)
		L	58 (40%)	−2.1 (3%)	0.82 (14%)
father	August	R	57 (29%)	−5.5 (81%)	0.99 (87%)
		L	56 (21%)	−5.0 (71%)	1.06 (96%)

The results marked in red were obtained post-treatment.

**Table 4 diagnostics-12-02701-t004:** Optical Coherence Tomography-thickness of Internal Limiting Membrane-Retinal Pigment Epithelium (ILM-RPE), Ganglion Cells Layer and Inner Plexiform Layer (GCL + IPL), Retinal Nerve Fiber Layer (RNFL); R—right eye; L—left eye.

PATIENT	EYE	MACULA		DISC
		ILM-RPE Thickness [µm]	Mean GCL + IPL Thickness [µm]	Mean RNFL Thickness [µm]
middle child	R	251	54	80
	L	247	52	75
eldest child	R	233	52	66
	L	233	53	67
youngest child	R	253	66	89
	L	254	64	81
mother	R	284	70	65
	L	287	67	65
father	R	270	82	91
	L	270	81	92

## Data Availability

Not applicable.

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
