# Peer review of "Phenotypic Variation of Autosomal Recessive Leber Hereditary Optic Neuropathy (arLHON) in One Family"

_diagnostics, 2022, doi:10.3390/diagnostics12112701_

Round 1

Reviewer 1 Report

Thank you for the opportunity to review this manuscript. The authors present an interesting paradigm with phenotypic variations of Leber's optic neuropathy in a family. The authors have presented the cases well but there are several issues in the manuscript that need to be addressed. Firstly, since this is a case series of a family, I would recommend taking out the year of births and calendar years from the manuscript. In writing case reports, efforts should be made to remove any identifiers unless absolutely necessary. In case of this manuscript, since multiple family members are presented, year of birth makes identification more plausible. Instead of calendar years, chronological years should be used in the manuscript. Secondly, there are multitude of grammatical errors in the manuscript, to the point that the line loses meaning ex. line 57. I would recommend reviewing the grammar thoroughly before resubmitting the manuscript. Lastly, I would recommend adding a paragraph describing the mechanism of action of Idebenone for treatment of Leber's optic neuropathy.

Author Response

Thank you very much for a valuable comments to the manuscript.

We took off year of births and calendar years to avoid patients’ identification.

We added a paragraph describing the mechanism of action of Idebenone for treatment of Leber optic neuropathy.

The manuscript was corrected by certified English translator.

Reviewer 2 Report

Pojda-Wilczek et al. realized a very interesting article describing the “Phenotypic variation of autosomal recessive Leber hereditary optic neuropathy (arLHON) in one family”. I consider the manuscript very interesting but, at the same time, I suggest several revisions needed to improve the reliability and the completeness of the paper: 

·      The “Introduction” section should be more detailed and informative of the pathology.

·      The “Discussion” section should be improved. I suggest to add perspectives about the role of coding and non-conding RNAs to improve the knowledge of molecular mechanisms involved into LHON pathology. Regarding this, I suggest to add the recent PMID: 32877751, PMID: 32184807 and doi: 10.3390/antiox11101967, which could represent a substrate able to enforce the role of considered cellular mechanisms.

·      Finally, manuscript requires serious English revisions and typos correction.

Author Response

Thank you very much for a valuable comments to the manuscript.

We added some comments about LHON pathology to the “Introduction”

As far as “Discussion” improvement is concerned we added some information about Idebenone.

We have read the recommended articles very carefully. We found them extremely interesting but not directly connected to our case study. We are sorry, but we are not able to cited them.

The manuscript was corrected by certified English translator.

Round 2

Reviewer 1 Report

Thank you for the opportunity to review this manuscript. My comments have been appropriately addressed by the authors.